# Work Situation of Midwives in Spain: Perception of Autonomy and Intention to Leave the Profession: A Cross-Sectional Study

**DOI:** 10.3390/healthcare12191994

**Published:** 2024-10-06

**Authors:** Susana Iglesias-Casás, Rafael Vila-Candel, Desirée Mena-Tudela, Anna Martín-Arribas, Fátima Leon-Larios

**Affiliations:** 1Primary Care Management of Tenerife, 38650 Tenerife, Spain; siglcas@gobiernodecanarias.org; 2Faculty of Health Sciences, Universidad Internacional de Valencia (VIU), 46002 Valencia, Spain; 3La Ribera Primary Health Department, 46600 Alzira, Spain; 4Foundation for the Promotion of Health and Biomedical Research in the Valencian Region (FISABIO), 46020 Valencia, Spain; 5PECAWOL (Perinatal care and Women’s Health), Joint Research Unit FISABIO-UJI, 46020 Valencia, Spain; dmena@uji.es; 6Nursing Department, Health Science Faculty, Instituto Feminista, Universitat Jaume I, 12071 Castellón, Spain; 7Ghenders Research Group, School of Health Sciences Blanquerna, Universitat Ramon Llull, 08025 Barcelona, Spain; annama7@blanquerna.url.edu; 8Nursing Department, School of Nursing, Physiotherapy and Podiatry, University of Seville, 41009 Seville, Spain; fatimaleon@us.es

**Keywords:** midwifery workforce, job autonomy, job satisfaction, quality of healthcare, workplace environment, continuity of care

## Abstract

Background: Developed countries report specific issues regarding the declining midwifery workforce, and their shortage could have serious consequences for women’s sexual and reproductive health. The aim was to understand the perception of autonomy among midwives working in Spain, as well as factors related to their intention to leave the profession and their work environment. Method: A descriptive and cross-sectional study using an online questionnaire. Population: midwives working in Spain in any field (clinical, research, teaching, or management). Results: A sample of 1060 midwives was obtained. Of these, 53.7% (*n* = 569) feel autonomous in their work, 92.4% (*n* = 978) perceive that their profession frequently suffers from external interference, 46.6% (*n* = 494) have experienced sexist behaviors at work, and 53% (*n* = 561) have considered leaving the profession in the last year. Midwives with less than 10 years of experience (57.7%), those aged 31–45 years (59.8%), those with temporary contracts (38.3%), and those working in hospital care (71.9%) show a higher rate of considering leaving the profession (*p* < 0.001). Conclusions: Considering the current midwifery workforce crisis in Spain, it seems urgent to improve the working conditions of midwives to ensure the continuity and quality of women’s sexual and reproductive healthcare.

## 1. Introduction

The midwife, as a professional figure of reference, contributes to the survival, improvement in health, and well-being of women and their newborns [1,2]. In 2023, the Bucharest Declaration called for political action and a commitment to protect, support, and invest in healthcare workers across Europe and Central Asia, as well as to address the pressing challenges facing this group [3,4]. Developed countries are reporting specific issues regarding the decline in the midwifery workforce or doubts about whether their numbers are sufficient to meet the current or future needs of the population [5,6,7].

The shortage of midwives could have serious consequences for the care of women during pregnancy, childbirth, and the early parenting period [8,9]. It has been documented that, in countries with low development indices, interventions provided by midwives through basic universal coverage could prevent 41% of maternal deaths, 39% of neonatal deaths, and 26% of stillbirths, which equates to 2.2 million avoidable deaths annually [10].

The International Confederation of Midwives (ICM) defined midwives’ autonomy as their ability to determine and control the standards for their own education, regulation, and practice, recognizing that in some countries, midwives face difficulties in this area due to medical hierarchies, government authorities seeking to control and limit the scope of their work, and the misuse of healthcare policies and protocols [11].

It has been recommended that midwives’ roles be expanded, along with an improvement in their working environment, support for their emotional well-being, adequate recognition and salaries, and their inclusion in policy development and leadership roles to ensure proper staffing of the profession [1]. Factors such as excessive workload staff shortages, inability to provide continuity of midwifery care, harassment, and lack of effective managerial support have been described as determinants of midwives’ emotional well-being [12,13]. There is a link between high levels of emotional distress and the intention to leave the profession [14]. Other obstacles, such as healthcare systems dominated by the medical model, lack of financial resources to adequately staff teams, and gender biases (the midwifery profession is highly feminized), prevent them from reaching their full potential [15].

In Spain, the competencies of midwives include a wide range of responsibilities focused on women’s sexual and reproductive health [16]. The current ratio is 6.1 midwives per 10,000 women aged 14–65, which is lower than the European median of 9.1. This low national ratio limits their availability to fully exercise all their professional competencies, meaning that the healthcare system cannot offer equitable and comprehensive sexual and reproductive healthcare. The Federation of Spanish Midwives (FAME) and the Spanish Midwives Association (AEM) have recently warned about the current shortage of midwives, as well as the lack of planning to ensure generational turnover, calling on the government to take an active and responsible role [16,17].

Given the current context, the main objective of this study was to understand the perception of autonomy among midwives working in Spain. As secondary objectives, we aimed to analyze the differences in their intention to leave the profession according to sociodemographic and professional variables and to explore certain aspects of their work environment.

## 2. Materials and Methods

### 2.1. Design

A descriptive, cross-sectional study was conducted using an online questionnaire. The questionnaire was hosted on a secure platform (Google Forms©) and was available from 7 February to 8 March 2023, after which it was closed to prevent further responses. Data collection allowed only one response per user, and participants were required to agree to the terms (purpose of the research, the voluntary nature of participation, and assurances regarding data privacy) to proceed before beginning the survey.

### 2.2. Study Population

Midwives practicing in Spain in any field (clinical, research, teaching, or management) agreed to participate in the study. The study was promoted by the Canary Islands Midwives Association (ACAMAT), which contacted all nursing colleges and midwifery associations in Spain via email, requesting their collaboration in disseminating the survey and explaining its objectives. The questionnaire was distributed via email and social media. The study population consisted of 8084 midwives actively working in Spain as of 31 December 2022 [16]. A total of 1060 midwives participated in the survey, representing 13.1% of the registered midwives in the country. The online survey was primarily disseminated through nursing colleges that chose to collaborate by sharing the survey link via their social media channels, encouraging their members to participate. The inclusion criteria required participants to be active midwives in Spain during the study period and to agree on the terms of survey. Exclusion criteria specified that the survey targeted only those currently practicing in Spain. To ensure data completeness, all survey items were mandatory, and incomplete responses were not accepted.

### 2.3. Variables and Data Collection Tool

An ad hoc electronic form was designed and organized into four distinct parts:(1)Sociodemographic and professional data of the participants:gender (woman/man), age (in full years), autonomous community of residence, years of work experience (in full years), healthcare sector (public healthcare/private healthcare), midwifery academic background (Resident Nurse Intern—RNI in Spain, non-Spanish university, or other), type of contract (interim, temporary, permanent, or self-employed), main practice (hospital care, primary care/ambulatory clinic, home birth/birthing center, or other: teaching, research, or management)(2)Perception of midwives’ autonomy, selected by the research team after reviewing the available literature on the subject [18,19,20,21]. The 25 items included in this section were rated on a Likert scale from 1 to 5 (5 strongly agree–1 strongly disagree).(3)Work environment evaluation: harassment (refers to whether the participant has experienced or witnessed workplace harassment), sexism (evaluates whether there are sexist and discriminatory attitudes, comments, or behaviors based on gender), questioning (refers to whether the worker feels that her work, decisions, or abilities are constantly doubted without justified reason), disrespect (examines whether the worker has been treated disrespectfully by superiors, colleagues, or subordinates), intrusiveness (refers to the presence of individuals performing tasks or roles for which they are not qualified, creating an inappropriate work environment). All these variables had dichotomous (yes/no) responses, and their definitions were provided to the participants.

Additionally, questions were included on perceived work environment: hostile (describes a competitive and conflict-prone environment), chaotic (indicates a disorganized environment with a lack of structure), depressing (describes an environment that demotivates and negatively affects employees’ mood), stressful (describes a high-pressure environment where employees feel constantly overwhelmed), cordial (reflects a friendly and collaborative work environment), motivating (refers to an environment that encourages employees to improve and achieve their goals).

Finally, there were questions about midwives’ feelings towards their workplaces: accomplished (feeling that he/she is meeting her goals and realizing her potential), invalidated (feeling his/her abilities are being ignored or suppressed), exhausted (experiencing feelings of burnout or saturation), recognized (perceiving that his/her work is valued by others), tired (feeling physically or mentally exhausted), content (experiencing a sense of well-being and satisfaction), fearful (feeling fear, whether for her safety, job stability, or due to workplace conflicts), angry (experiencing anger or frustration due to workplace situations), burnt out (showing signs of emotional or physical exhaustion from work), satisfied (feeling satisfaction with her performance and outcomes), powerless (feeling he/she has no control or ability to influence workplace situations). For each feeling, participants could select from five categories (very little, little, quite a lot, a lot, always).

In January 2023, a pilot survey was conducted with 10 participants, all midwives with over 15 years of work experience. The evaluation was positive in terms of item comprehension and acceptability, and no changes were made. Participants reported that it took about 5 min to complete the survey.

### 2.4. Data Analysis

Descriptive statistics were used to analyze the characteristics of the sample, with the mean and standard deviation (SD) applied for quantitative variables and ranges and percentages for qualitative variables. Some variables were categorized to facilitate analysis and comprehension of the results, such as work experience (<10 years/10–19 years/>20 years) and age (<30/30–44/45–65). To analyze the participants’ expressed intention to leave the profession, responses were grouped into two categories: Yes (agree/strongly agree) and No (neither agree nor disagree/disagree/strongly disagree).

To assess the reliability of the questions used in the study, two internal consistency coefficients were calculated: McDonald’s Omega and Cronbach’s Alpha. The Chi-square test (χ^2^) was used to describe the distribution of midwives’ experiences and the possibility of leaving the profession according to their sociodemographic variables.

Data analysis was performed using the statistical software package SPSS v.28.0 (IBM Corp. 2018. IBM SPSS Statistics for Windows, Armonk, NY, USA). A *p*-value < 0.05 was considered statistically significant.

### 2.5. Ethical Considerations

At the beginning of the form, participants were informed about the study’s purpose, and it was made clear that by accessing and completing the form, they were consenting to the use of the data collected. Anonymity, voluntary participation, and the proper use of the data were guaranteed. Contact details for the research team were provided.

## 3. Results

The total sample analyzed consisted of 1060 midwives. The study sample was predominantly composed of 95.1% (*n* = 1008) women. The average age of the sample was 39.48 (9.4) years, with an average work experience of 12.33 (9.2) years. Regarding academic qualifications, 87.9% (*n* = 932) of the participants held the RNI (Resident Nurse Intern) qualification from Spain. Concerning contract type, 60.1% (*n* = 637) had interim or temporary contracts. In terms of activities performed in the past year, 61.6% (*n* = 653) of participants were engaged in specialized care. The sociodemographic characteristics are shown in Table 1.

Table 2 presents the descriptive results regarding midwives’ perceptions of autonomy. In relation to the perception of autonomy, 92.7% (*n* = 982) of midwives agree or strongly agree that they advocate for sexual and reproductive rights. A further 96% (*n* = 1018) agree or strongly agree that they promote natural childbirth. On the other hand, only 22.8% (*n* = 242) agree or strongly agree that the team is led by midwives, and a mere 17.2% (*n* = 182) believe that their managers value their work. Regarding Continuing Education and Autonomy, 49.2% (*n* = 522) agree or strongly agree that they receive adequate continuing education, and 53.7% (*n* = 569) feel autonomous in their work. Concerning workload and work–life balance, 63.6% (*n* = 674) disagree or strongly disagree that the workload is appropriate, and 46.3% (*n* = 491) feel they can effectively balance their personal and professional lives. Lastly, regarding the intention to leave the profession, 53% (*n* = 561) have considered leaving the profession in the last year. The reliability analysis showed an Omega coefficient of 0.88 and a Cronbach’s Alpha of 0.89, indicating good internal consistency.

Table 3 describes the work situations affecting the professional environment of the surveyed midwives. Regarding professional intrusion, 92.4% (*n* = 978) of the midwives perceive that their profession frequently suffers from intrusion. Moreover, 46.6% (*n* = 494) have experienced sexist behaviors, and 82.5% (*n* = 875) have encountered classist attitudes. A total of 13.8% (*n* = 146) report having suffered harassment, and 26.9% (*n* = 285) of the respondents feel that they are treated with disrespect. Also, 51.1% (*n* = 541) of the midwives feel that their professional opinions are questioned.

Table 4 shows significant positive feelings and perceptions about the work environment, such as feeling accomplished, recognized, happy, and satisfied, alongside notable concerns about stress, exhaustion, and a perception of a hostile and chaotic environment. Positive perceptions are significant, with 53.2% (*n* = 564) finding the environment quite cordial and 46.3% (*n* = 491) feeling quite accomplished. In contrast, negative perceptions and feelings are also prominent, with 40.3% (*n* = 427) finding the environment quite stressful and 35.5% (*n* = 376) feeling quite exhausted.

Table 5 shows a comparative analysis of midwives’ experiences regarding their sociodemographic and occupational variables. Regarding professional intrusion, there are statistically significant differences related to age, with a higher perception of intrusion in the 31–45 years group (*p* < 0.001). Additionally, men perceive less intrusion compared to women (*p* = 0.031).

In terms of sexism, significant differences were found, with a higher perception of sexism in the groups with less than 10 years and 10-20 years of experience (*p* < 0.001), in the 31–45 years group (*p* < 0.001), and in midwives with temporary contracts (*p* < 0.001).

Regarding classism, a higher perception is observed in the groups with less than 10 years and more than 20 years of experience (*p* < 0.001), in the 31–45 years and >46 years groups (*p* < 0.001), in midwives with temporary contracts (*p* < 0.001), and in those who have worked in primary care in the past year (*p* < 0.001).

For harassment, a higher incidence is perceived in the 31–45 years group (*p* < 0.001), in temporary contracts (*p* = 0.002), in public health (*p* = 0.017), and in hospital care (*p* = 0.009).

Finally, the questioning variable shows statistically significant differences in the groups with less than 10 years and more than 20 years of experience (*p* < 0.001), in the <=30 years and 31–45 years groups (*p* < 0.001), in midwives with temporary contracts (*p* < 0.001), and in those working in hospital care (*p* < 0.001).

The results of the comparative analysis regarding the intention to leave the profession. Midwives with less than 10 years of experience (57.7%), those in the 31–45 age group (59.8%), those with temporary contracts (38.3%), and those working in hospital care (71.9%) show the highest rates of considering leaving the profession (*p* < 0.001).

## 4. Discussion

The purpose of this study was to understand the level of autonomy perceived by midwives working in Spain, as well as to analyze the variables associated with the development of autonomy.

Previous studies have highlighted that midwives’ autonomy remains a vague and difficult-to-define concept at a general level [19], and it is determined by the legal, cultural, and professional context of each country [20]. In the most recent definition of the term, five key themes were identified: adequate education, competence, experience, quality care, and collaboration with stakeholders, emphasizing the need for interprofessional education to strengthen midwives’ collaboration and autonomy [21]. However, this term was not defined or studied in Spain until 2023, establishing it as a novel area that requires further research [22].

Regarding the professional profile of the study participants, we found professionals primarily trained in Spain with an average of 10 years of professional experience and job stability. These aspects may contribute to a deeper analysis of the reality of midwives’ autonomy. Similar to studies conducted in other countries, midwives are identified as well-trained, confident, and competent in their professional role, especially when given the opportunity to work in a supportive environment [23,24,25].

The workplace, whether hospital or primary care, can significantly influence midwives’ perception of autonomy. In this study, we found that autonomy tends to be lower in hospital settings, a result similar to other authors [26,27].

This study also identified that interprofessional hierarchical relationships constitute a significant barrier to achieving professional autonomy. Midwives who participated in this study identified areas for improvement in management support and collaboration with other professional groups, feeling inadequately supported by them [19,26,28]. Thus, supportive relationships and quality leadership are perceived as empowering and promote professional development. Therefore, well-implemented supervision or coordination, aimed at professional support rather than rigorous monitoring and control of employees’ actions, can improve both the professional and personal development of midwives [29,30].

The results of this study show that midwives identified the need to be more active in developing protocols, as reflected in their competencies [4]. They are not always included in working groups established to develop these protocols, and their participation is often limited. However, when midwives are included in participation, planning, and decision-making processes, both clinical practice and professional satisfaction improve [30,31].

Among the barriers identified for optimal professional development of midwives are working conditions, family reconciliation, and perceived professional intrusion. These barriers have a significant impact on midwives’ daily practice and their perceptions of autonomy [3]. For example, precarious working conditions, such as a lack of resources and staff, force midwives to take on excessive workloads, which not only affects their physical and mental well-being but also limits their ability to practice autonomously, be woman-centered, and provide a continuum of care [13]. This situation is consistent with findings from other studies [5,6], which observe how work–life balance is a constant concern for midwives, affecting job satisfaction and commitment to the profession.

To gain deeper insights into how these specific barriers affect midwives’ daily practice, it is crucial to consider direct examples of their experiences. The results of this study showed that lack of support and recognition from supervisors limits the ability to make autonomous decisions in childbirth care, forcing adherence to rigid protocols that do not always meet the individual needs of women [32,33]. On the other hand, professional intrusion resulting from a shortage of midwives [16] prevents them from fully developing their competencies, negatively affecting their confidence and professional satisfaction [34,35], diluting the quality of care, creating interprofessional conflicts, and affecting the work environment. More studies are needed to determine this finding in Spain.

More than half of the study participants with less than 10 years of work experience have considered leaving the profession at some point; a similar percentage was found in Australia [32]. It is observed that with greater professional experience, there is less desire to leave the profession. Previous studies have identified causes such as sexism, classism, and lack of respect, which contribute to an unfavorable work environment for the professional development of midwives [32]. These findings are consistent with the results of our research. Additionally, a lack of staff and resources has been identified as a significant stress factor contributing to an unhealthy work environment [12].

Job precariousness has also been highlighted in this research as a determinant of work burnout, with midwives experiencing greater job instability being more likely to question their continued presence in the profession [32,34]. Age and professional experience seem to influence the perception of situations involving classism, sexism, professional intrusion, and questioning of their professional opinions. Midwives with more years of work experience feel more frustrated by the lack of recognition and support, influenced by power and gender dynamics in the workplace. Conversely, younger midwives are more likely to feel overwhelmed and less recognized, leading them to question their competence more easily [7]. Further research is needed on these findings.

The study demonstrated high internal consistency for the ad hoc survey used to assess midwives’ perceptions of autonomy in Spain, as evidenced by a robust Omega coefficient and Cronbach’s Alpha. This suggests that the survey items were cohesive and effectively measured the construct of autonomy. The Perception of Empowerment in Midwifery Scale (PEMS) originally developed in 2007 [36] and revised in 2015 (PEMS-R) [37], has been adapted to various languages, including Italian, Portuguese, and Persian [38,39,40]. However, the lack of a similar tool in Spanish has posed challenges in assessing midwives’ empowerment within the Spanish context. After our data collection was completed, González-de la Torre et al. translated and validated the PEMS scale into Spanish [22]. However, this version was not utilized in our study, as our research had already concluded by that time. It is important to note that our primary aim was not to translate, culturally adapt, or validate the PEMS or PEMS-R for Spanish midwives. Instead, our study aimed to provide a more comprehensive understanding of autonomy by incorporating additional variables specific to the Spanish midwifery context.

Among the limitations of this study is the potential for response bias due to voluntary or motivated participation, as midwives with more polarized experiences in the profession, whether positive or negative, may be more interested in completing the questionnaire. Those with more neutral or less intense experiences are less likely to respond [8]. The responses provided by participants may not be representative of the midwives’ community in Spain, cannot be generalized due to voluntary participation, and should be interpreted with caution. Additionally, we did not perform prior power calculations, which is particularly important in exploratory research. At the same time, the study represents an estimated response rate of 13.1% of the target population of active midwives in Spain; the lack of a defined effect size prior to data collection may limit the generalizability of our findings. Future research should aim to incorporate power analyses to enhance the robustness of the results and establish more definitive conclusions regarding the perceptions of autonomy among midwives in Spain.

A strength of the study is the large number of participants and the geographical diversity of the responses, which allows for a general analysis and comparison between different areas of the country. While the high internal consistency supports the validity of the findings, the ad hoc nature of the survey may limit the generalizability of the results. This study allows for further advancement and deepening in the field of professional autonomy of midwives in Spain and the reasons that may lead midwives to leave the profession.

## 5. Conclusions

This study has achieved its main objective by providing a comprehensive view of the perception of autonomy among midwives working in Spain and their intention to leave the profession. Differences in the intention to leave reveal that midwives with more experience feel more frustrated by the lack of recognition, while younger midwives feel more questioned.

Given the current crisis in the profession regarding the workforce, it seems urgent to improve the working conditions of midwives in our country. The Ministry of Health should protect women’s sexual and reproductive health by investing in this healthcare resource, implementing measures not only to improve the midwife-to-woman ratio but also to prevent midwives from leaving the profession, such as increasing their autonomy and improving their working conditions and environment. Explicit institutional support in developing their competencies, considering changes in the training model, providing efficient management, and encouraging their active participation in the development of sexual and reproductive health policies and protocols may be key to the present and future of this group.

## Figures and Tables

**Table 1 healthcare-12-01994-t001:** Midwives’ sociodemographic characteristics (*n* = 1060).

Variable		*n* (%)
Work Experience	<10 years	495 (46.7)
	10–19 years	342 (32.3)
	>20 years	223 (21.0)
Age	<30 years	196 (18.5)
	30–44 years	587 (55.4)
	45–65 years	274 (25.8)
	Missing	3 (0.3)
Sex	Men	52 (4.9)
	Women	1008 (95.1)
Midwifery Academic Background	RNI Spain	932 (87.9)
	Non-Spanish University	97 (9.2)
	Other	31 (2.9)
Contract Type	Interim	319 (30.1)
	Temporary	318 (30.0)
	Permanent	403 (38.0)
	Self-employed	20 (1.9)
Healthcare sector	Public healthcare	1013 (95.6)
	Private healthcare	47 (4.4)
Main Practice	Hospital care	653 (61.6)
	Primary care/ambulatory clinic	378 (35.7)
	Home birth/birthing center	11 (1.0)
	Other (teaching, research, management)	18 (1.7)

RNI: Resident Nurse Intern (Spain).

**Table 2 healthcare-12-01994-t002:** Midwives’ perception of autonomy (*n* = 1060).

	Strongly Disagree	Disagree	Neither Agree nor Disagree	Agree	Strongly Agree
Item	*n* (%)	*n* (%)	*n* (%)	*n* (%)	*n* (%)
In my work, I advocate for women’s sexual and reproductive rights	25 (2.4)	14 (1.3)	39 (3.7)	374 (35.3)	608 (57.4)
One goal of my profession is to promote the physiological process of childbirth	22 (2.1)	7 (0.7)	14 (1.3)	220 (20.8)	707 (75.2)
I work in a team led by midwives	315 (29.7)	339 (32.0)	164 (15.5)	155 (14.6)	87 (8.2)
I have the adequate training to develop my profession at all levels (care, teaching, and research)	30 (2.8)	179 (16.9)	152 (14.3)	440 (41.5)	259 (24.4)
I have the skills and knowledge necessary to perform my role	14 (1.3)	12 (1.1)	33 (3.1)	494 (46.6)	507 (47.8)
Managers value me and recognize my contribution to the sexual and reproductive health care of women	301 (28.4)	344 (32.5)	233 (22.0)	146 (13.8)	36 (3.4)
I have the support and endorsement of my management	238 (22.5)	316 (29.8)	300 (28.3)	168 (15.8)	38 (3.6)
I have access to the necessary equipment and resources to work with quality	78 (7.4)	271 (25.6)	229 (21.6)	414 (39.1)	68 (6.4)
I have adequate access to continuing education resources	78 (7.4)	244 (23.0)	217 (20.5)	427 (40.3)	94 (8.9)
I am recognized as a professional by the medical team	56 (5.3)	143 (13.5)	204 (19.2)	511 (48.2)	146 (13.8)
The medical team recognizes my contribution to the sexual and reproductive health care of women	102 (9.6)	271 (25.6)	277 (26.1)	344 (32.5)	66 (6.2)
In my work, there is a hierarchical relationship of subordination of midwives to medical professionals	35 (3.3)	164 (15.5)	206 (19.4)	403 (38.0)	252 (23.8)
I have the support of my colleagues in the multidisciplinary team	43 (4.1)	182 (17.2)	312 (29.4)	426 (40.2)	97 (9.2)
In my work, I am able to refuse to perform practices I consider unnecessary or harmful	41 (3.9)	251 (23.7)	240 (22.6)	414 (39.1)	114 (10.8)
I am autonomous in my work	48 (4.5)	207 (19.5)	236 (22.3)	450 (42.5)	119 (11.2)
I feel comfortable with the level of responsibility required by my job	41 (3.9)	168 (15.8)	132 (12.5)	527 (49.7)	192 (18.1)
I am asked to participate in creating protocols that I will use in my daily work	147 (13.9)	262 (24.7)	204 (19.2)	321 (30.3)	126 (11.9)
In my job, I apply protocols based on high-quality scientific evidence that are periodically reviewed	106 (10.0)	217 (20.5)	195 (18.4)	388 (36.6)	154 (14.5)
Conflicts between the medical model and the midwifery model in caring for women are common in my job	55 (5.2)	233 (22.0)	256 (24.2)	322 (30.4)	194 (18.3)
There are enough midwives in my job, and the workload allows me to perform my duties with quality	299 (28.2)	375 (35.4)	121 (11.4)	213 (20.1)	52 (4.9)
My job allows me to satisfactorily balance personal and professional life	121 (11.4)	241 (22.7)	207 (19.5)	406 (38.3)	85 (8.0)
My rights in case of pregnancy, maternal breastfeeding, maternity, or paternity are effectively protected in my job	70 (6.6)	159 (15.0)	224 (21.1)	458 (43.2)	149 (14.1)
My working conditions (salary, breaks, type of contract, workload) are dignified	261 (24.6)	373 (35.2)	149 (14.1)	238 (22.5)	39 (3.7)
My profession is known and respected by society	67 (6.3)	328 (30.9)	260 (24.5)	360 (34.0)	45 (4.2)
Considered leaving in last year	292 (27.5)	270 (25.5)	132 (12.5)	209 (19.7)	157 (14.8)

**Table 3 healthcare-12-01994-t003:** Work situations experienced by midwives (*n* = 1060).

Situation	Yes *n* (%)	No *n* (%)
My profession frequently suffers from intrusion	979 (92.4)	81 (7.6)
I have experienced sexist behaviors	494 (46.6)	566 (53.4)
I have experienced classist attitudes	874 (82.5)	186 (17.5)
I have suffered harassment	146 (13.8)	914 (86.2)
I am treated with disrespect	285 (26.9)	775 (73.1)
My professional opinion is questioned	542 (51.1)	518 (48.9)

**Table 4 healthcare-12-01994-t004:** Questions about work environment (*n* = 1060).

		Very Little *n* (%)	Little *n* (%)	Quite a Lot *n* (%)	Much *n* (%)	Always *n* (%)
Work Environment	Hostile	440 (41.5)	395 (37.3)	158 (14.9)	54 (5.1)	13 (1.2)
	Chaotic	273 (25.8)	466 (44.0)	245 (23.1)	64 (6.0)	12 (1.1)
	Cordial	36 (3.4)	168 (15.8)	564 (53.2)	234 (22.1)	58 (5.5)
	Stressful	102 (9.6)	337 (31.8)	427 (40.3)	154 (14.5)	40 (3.8)
	Motivating	135 (12.7)	433 (40.8)	348 (32.8)	116 (10.9)	28 (2.6)
	Depressing	436 (41.1)	381 (35.9)	171 (16.1)	57 (5.4)	15 (1.4)
Daily Feelings	Accomplished	61 (5.8)	206 (19.4)	491 (46.3)	251 (23.7)	51 (4.8)
	Overlooked	415 (39.2)	424 (40.0)	169 (15.9)	44 (4.2)	8 (0.8)
	Fed Up	205 (19.3)	385 (36.3)	292 (27.5)	150 (14.2)	28 (2.6)
	Recognized	119 (11.2)	406 (38.3)	392 (37.0)	120 (11.3)	23 (2.2)
	Exhausted	105 (9.9)	327 (30.8)	376 (35.5)	204 (19.2)	48 (4.5)
	Happy	38 (3.6)	243 (22.9)	475 (44.8)	251 (23.7)	53 (5.0)
	Fearful	412 (40.0)	424 (40.0)	168 (15.8)	47 (4.4)	9 (0.8)
	Angry	249 (23.5)	421 (39.7)	242 (22.8)	126 (11.9)	22 (2.1)
	Burned Out	210 (19.8)	381 (35.9)	254 (24.0)	171 (16.1)	44 (4.2)
	Satisfied	56 (5.3)	290 (27.4)	435 (41.0)	222 (20.9)	57 (5.4)
	Powerless	179 (16.9)	335 (31.6)	272 (25.7)	224 (21.1)	50 (4.7)

**Table 5 healthcare-12-01994-t005:** Distribution of midwives’ experiences and consideration of leaving the profession by sociodemographic variables (*n* = 1060).

	Intrusiveness			Sexism			Classism			Harassment			Disrespect			Questioning			Abandonment		
	No	Yes		No	Yes		No	Yes		No	Yes		No	Yes		No	Yes		No	Yes	
Variable	*n* (%)	*n* (%)	*p* *	*n* (%)	*n* (%)	*p* *	*n* (%)	*n* (%)	*p* *	*n* (%)	*n* (%)	*p* *	*n* (%)	*n* (%)	*p* *	*n* (%)	*n* (%)	*p* *	*n* (%)	*n* (%)	*p* *
Work Experience																					
<10 years	30 (37.0)	465 (47.5)	0.186	244 (43.1)	251 (50.8)	<0.001	61 (32.8)	434 (49.7)	<0.001	435 (47.6)	60 (41.1)	0.212	349 (45.0)	146 (51.2)	0.002	191 (46.2)	116 (37.3)	<0.001	284 (40.9)	211 (57.7)	<0.001
10–19 years	30 (37.0)	312 (31.9)		178 (31.4)	164 (33.2)		61 (32.8)	281 (32.2)		286 (31.3)	56 (38.4)		242 (31.2)	100 (35.1)		135 (32.7)	104 (33.4)		239 (34.4)	103 (28.1)	
>20 years	21 (26.0)	202 (20.6)		144 (25.4)	79 (16.0)		64 (34.4)	159 (18.2)		193 (21.1)	30 (20.5)		184 823.7)	39 (13.7)		87 (21.1)	91 (29.3)		171 (24.6)	52 (14.2)	
Age																					
<30 years	13 (16.0)	183 (19.0)	<0.001	93 (16.5)	103 (20.9)	<0.001	16 (8.7)	180 (20.6)	<0.001	176 (19.3)	20 (13.7)	0.268	138 (17.9)	58 (20.4)	0.028	84 (20.3)	35 (11.3)	<0.001	117 (16.9)	79 (21.6)	<0.001
30–44 years	32 (39.5)	555 (57.0)		295 (52.4)	292 (59.1)		88 (47.8)	499 (57.2)		501 (55.0)	86 (58.9)		417 (54.0)	170 (59.6)		218 (52.8)	168 (54.4)		368 (55.3)	219 (59.8)	
45–65 years	36 (4.5)	238 (24.0)		175 (31.1)	99 (20-0)		80 (43.5)	194 (22.2)		234 (25.7)	40 (27.4)		217 (28.1)	57 (20.0)		111 (26.9)	106 (34.3)		206 (29.8)	68 (18.6)	
Sex																					
Men	8 (9.9)	44 (4.5)	0.031	29 (5.1)	23 (4.7)	0.725	14 (7.5)	38 (4.3)	0.068	41 (4.5)	11 (7.5)	0.113	34 (4.4)	18 (6.3)	0.197	16 (3.9)	16 (5.1)	0.413	31 (4.5)	21 (5.7)	0.362
Women	73 (90.1)	935 (95.5)		537 (94.9)	471 (95.3)		172 (92.5)	836 (95.7)		873 (95.5)	135 (92.5)		741 (95.6)	267 (93.7)		397 (96.1)	295 (94.9)		663 (95.5)	345 (94.3)	
Contract Type																					
Interim	27 (33.3)	292 (29.8)	0.197	182 (32.2)	137 (27.7)	<0.001	65 (34.9)	254 (29.1)	<0.001	279 (30.5)	40 (27.4)	0.002	239 (30.8)	80 (28.1)	0.153	128 (31.0)	98 (31.5)	<0.001	199 (28.7)	120 (32.8)	<0.001
Temporary	18 (22.3)	300 (30.6)		139 (24.6)	179 (36.2)		33 (17.7)	285 (32.6)		282 (30.9)	36 (24.7)		221 (28.5)	97 (34.0)		127 (30.8)	61 (19.6)		178 (25.6)	140 (38.3)	
Permanent	36 (44.4)	367 (37.6)		236 (41.7)	167 (33.8)		82 (44.1)	321 (36.7)		341 (37.3)	62 (42.5)		303 (39.1)	100 (35.1)		152 (36.8)	147 (47.3)		308 (44.4)	95 (26.0)	
Self-employed	0 (0.0)	20 (2.0)		9 (1.6)	11 (2.2)		6 (3.2)	14 (1.6)		12 (1.3)	8 (5.5)		12 (1.5)	8 (2.8)		6 (1.5)	5 (1.6)		9 (1.3)	11 (3.0)	
Main Professional Field																					
Public healthcare	3 (3.7)	44 (4.5)	0.74	20 (3.5)	27 (5.5)	0.127	12 (6.5)	35 (4.0)	0.141	35 (3.8)	12 (8.2)	0.017	32 (4.1)	15 (5.3)	0.426	17 (4.1)	11 (3.5)	0.392	21 (3.0)	26 (7.1)	0.002
Private healthcare	78 (96.3)	935 (95.5)		546 (96.5)	467 (64.5)		174 (93.5)	839 (96.0)		879 (96.2)	134 (91.8)		743 (95.9)	270 (94.7)		396 (95.9)	300 (96.5)		673 (97.0)	340 (92.9)	
Main Practice																					
Home Birth Care	1 (1.2)	8 (0.8)	0.561	4 (0.7)	5 (1.0)	0.064	3 (1.6)	6 (0.7)	<0.001	5 (0.5)	4 (2.7)	0.009	6 (0.8)	3 (1.1)	<0.001	2 (0.5)	4 (1.3)	<0.001	6 (0.9)	3 (0.8)	<0.001
Hospital Care	42 (52.0)	611 (62.4)		329 (58.1)	324 (65.6)		80 (43.0)	573 (65.6)		551 (60.3)	102 (69.9)		447 (57.7)	206 (72.3)		223 (54.0)	173 (55.6)		390 (56.2)	263 (71.9)	
Primary Care/Ambulatory Clinic	36 (44.4)	342 (34.9)		222 (39.2)	156 (31.6)		97 (52.2)	281 (32.2)		340 (37.2)	38 (26.0)		305 (39.4)	73 (25.6)		180 (43.6)	126 (4.5)		283 (40.8)	95 (26.0)	
Birth Center	0 (0.0)	2 (0.1)		0 (0.0)	2 (0.4)		1 (0.5)	1 (0.1)		2 (0.2)	0 (0.0)		2 (0.3)	0 (0.0)		0 (0.0)	1 (0.3)		2 (0.3)	0 (0.0)	
Teaching/Research	1 (1.2)	7 (0.7)		4 (0.7)	4 (0.8)		2 (1.1)	6 (0.7)		8 (0.9)	0 (0.0)		8 (1.0)	0 (0.0)		6 (1.5)	2 (0.6)		6 (0.9)	2 (0.5)	
Other	1 (1.2)	9 (0.9)		7 (1.2)	3 (0.6)		3 (1.6)	7 (0.8)		8 (0.9)	2 (1.4)		7 (0.9)	3 (1.1)		2 (0.5)	5 (1.6)		7 (1.0)	3 (0.8)	

* Chi-squared test.

## Data Availability

All necessary data are supplied and available in the manuscript; however, the corresponding author will provide the dataset upon request.

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
