# Peer review of "Work Situation of Midwives in Spain: Perception of Autonomy and Intention to Leave the Profession: A Cross-Sectional Study"

_healthcare, 2024, doi:10.3390/healthcare12191994_

Round 1

Reviewer 1 Report

Comments and Suggestions for Authors

Summary

From perspective of workforce building, the article studied the perception of autonomy among midwives working in Spain, and factors related to their intention to leave the profession and their work. Its a meaningful topic in the context of protecting health workers, considering the declining midwifery workforce and their shortage across developed countries.

Specific comments:

1. Sampling methods: please specify the sampling process, including the size of target population, inclusion criteria, exclusion criteria, and response rate.

2. Methodology of data analysis: This manuscript only uses descriptive statistics to analyze the characteristics of the sample, however, the data value and its policy implications have not been fully presented. Therefore, it is recommended to conduct further analysis based on the findings of descriptive statistics, such as single-factor analysis or multi-factor analysis, to explore the underlying dimensions that explain the relationships between the multiple variables/items.

3. Table layout: The contents of Table 1, Table 5 and Table 6 are repeated and can be merged into one table by deleting the redundant parts. For example, please delete all columns of No (n %) and all columns of P*; then, please mark the p value with an asterisk (*) for indicating statistical significance when necessary, instead of showing all the detailed P value number.     

4. Workplace violence analysis: The manuscript listed workplace violence as one of the key words, however, no a single word of violence is mentioned in the main body. It appears that the concept of workplace violence was replaced by the items of work environment evaluation at the part of Materials and Methods (such as, harassment, sexism, disrespect), and the items of work situations experienced at the part of Results. Therefore, I recommend to introduce the link of the key word in the main body, to interpret the policy implication under the concept of workplace violence and harassment.

Author Response

Response to Reviewer 1 Comments

Thank you for your very positive and constructive feedback on our manuscript.

We have considered all your comments and suggestions and the comments made by the other reviewers attempting to improve/refine the original manuscript.

Below, you will find a point-by-point response to your comments (in red).

From perspective of workforce building, the article studied the perception of autonomy among midwives working in Spain, and factors related to their intention to leave the profession and their work. It’s a meaningful topic in the context of protecting health workers, considering the declining midwifery workforce and their shortage across developed countries.

Specific comments:

Point 1: Sampling methods: please specify the sampling process, including the size of target population, inclusion criteria, exclusion criteria, and response rate.

Response 1: Thank you for your comment. A new paragraph has been included in lines 98-106, page 3.

The study population consisted of 8084 midwives actively working in Spain, as of December 31, 2022 [16]. A total of 1060 midwives participated in the survey, representing 13.1% of the registered midwives in the country. The online survey was primarily disseminated through nursing colleges that chose to collaborate by sharing the survey link via their social media channels, encouraging their members to participate. The inclusion criteria required participants to be active midwives in Spain during the study period and to agree the terms of survey. Exclusion criteria specified that the survey targeted only those currently practicing in Spain. To ensure data completeness, all survey items were mandatory, and incomplete responses were not accepted. 

Point 2: Methodology of data analysis: This manuscript only uses descriptive statistics to analyze the characteristics of the sample, however, the data value and its policy implications have not been fully presented. Therefore, it is recommended to conduct further analysis based on the findings of descriptive statistics, such as single-factor analysis or multi-factor analysis, to explore the underlying dimensions that explain the relationships between the multiple variables/items.

Response 2: We appreciate your insightful feedback on our study. Our aim in this work was not to validate a scale or its translation or back-translation process. Nor its cultural or contextual adaptation. However, during the process of data analysis, we estimated Barlett's (<0.001) and KMO (0.901) which invited factor analysis. The results after principal component analysis showed a monofactorial scale that did not provide much information. Therefore, for methodological consistency, we calculated internal consistency in order to statistically assess whether the items measured the same underlying theoretical construct. The results were positive and are shown in the paper, in this way.

The aim of our study is to highlight the perception of autonomy among midwives practicing in Spain, emphasizing the latent risk of leaving the profession. The significance of this study lies in being the first national-level research on this topic, making it a starting point for future investigations. Moreover, its findings align with international studies, despite the inherent differences in healthcare systems across countries.

Point 3: Table layout: The contents of Table 1, Table 5 and Table 6 are repeated and can be merged into one table by deleting the redundant parts. For example, please delete all columns of No (n %) and all columns of P*; then, please mark the p value with an asterisk (*) for indicating statistical significance when necessary, instead of showing all the detailed P value number.     

Response 3: Thank you for your input. We have fuionated atable 5 and 6 in order to clarify this comment, in pages 9 and 10.

Point 4: Workplace violence analysis: The manuscript listed “workplace violence” as one of the key words, however, no a single word of “violence” is mentioned in the main body. It appears that the concept of “workplace violence” was replaced by the items of “work environment evaluation” at the part of Materials and Methods (such as, harassment, sexism, disrespect), and the items of “work situations experienced” at the part of Results. Therefore, I recommend to introduce the link of the key word in the main body, to interpret the policy implication under the concept of workplace violence and harassment.

Response 4: Thank you for your valuable suggestion. In line with your recommendation, we have opted to replace the keyword "workplace violence" with "work environment," as we believe this term more accurately captures our aim of understanding the conditions under which midwives work in Spain.

Reviewer 2 Report

Comments and Suggestions for Authors

Thank you for your paper, and I enjoyed reading about your well-designed research. I wish you every success in your ongoing work.

I have a couple of very minor suggested clarifications.

The introduction was clear, as were the materials and methods. My only comment re the methods is that for future work, I would caution against binary Yes/No options, as I think most people would tend to answer in the ‘sometimes’ arena. I appreciate that binary options can give a stronger result, however, including a range (such as with the rest of your survey) can give a more nuanced insight, and this would be especially valuable when investigating sensitive topics such as harassment.

In the description of the survey, you defined the various terms for the reader. Were these terms also defined for the study participants?

The sample seems to be a good size. I understand that it would be difficult to quantify the response rate because of the recruitment methods, however, did you do any power calculations? Do you know (approximately) what your total sample was, so that you could estimate the response rate?

In the Data Analysis section, can you include a statement on how you managed missing data or incomplete questionnaires? If there were no missing data or incomplete questionnaire, then confirm this.

In the Ethics section – I can see the ethics statement at the end of the work, but I was not clear regarding whether or not you formally gained ethical approval for the study, Could you please confirm within the text?

Results: The tables are well laid out and very clear.

The discussion was relevant to the results and linked to the wider literature.

Editing errors: Title – Sectional

Line 276/277: check sentence structure. Perhaps include ‘be’ – to practice autonomously, be women-centered…

Author Response

Response to Reviewer 2 Comments

Thank you for your very positive and constructive feedback on our manuscript.

We have considered all your comments and suggestions and the comments made by the other reviewers attempting to improve/refine the original manuscript.

Below, you will find a point-by-point response to your comments (in red).

Thank you for your paper, and I enjoyed reading about your well-designed research. I wish you every success in your ongoing work. I have a couple of very minor suggested clarifications.

Specific comments:

Point 1: The introduction was clear, as were the materials and methods. My only comment re the methods is that for future work, I would caution against binary Yes/No options, as I think most people would tend to answer in the ‘sometimes’ arena. I appreciate that binary options can give a stronger result, however, including a range (such as with the rest of your survey) can give a more nuanced insight, and this would be especially valuable when investigating sensitive topics such as harassment.

Response 1: We appreciate this valuable suggestion for future research. Indeed, when dealing with sensitive topics such as harassment, a broader range of options could provide a more detailed view of participants' experiences. Although in this study we opted for binary options to simplify the categorization of responses, we will take your recommendation into account for future work, aiming to better capture variations in experiences and perceptions.

Point 2: In the description of the survey, you defined the various terms for the reader. Were these terms also defined for the study participants?

Response 2: Thank you for your comment. Yes, the terms used in the survey, such as harassment, sexism, questioning, disrespect or intrusiveness, and questions on perceived work environment, were defined for the participants at the beginning of the relevant section of the questionnaire. This ensured that all respondents clearly understood the meaning of each item and could respond in an informed and accurate manner. We will add a note in the manuscript clarifying that the definitions were provided to the participants, in line 130 page 3.

Point 3: The sample seems to be a good size. I understand that it would be difficult to quantify the response rate because of the recruitment methods, however, did you do any power calculations? Do you know (approximately) what your total sample was, so that you could estimate the response rate?

Response 3: Thank you for your input. We appreciate your observation regarding the sample size. While we acknowledge that the sample of 1060 midwives who participated in the survey is significant, we did not perform prior power calculations due to the exploratory nature of our study and the lack of a defined effect size in this context.

The target population of active midwives in Spain is approximately 8084, and the estimated response rate was 13.1%. This value reflects participation in the study, although it is true that the recruitment methods make it difficult to quantify the exact response rate.

Since this is the first national study addressing the perception of autonomy among midwives in Spain, we believe our findings can serve as a starting point for future, more detailed research. In future studies, power analyses could be considered using effect sizes based on the results obtained.

We have included a new paragraph in lines 98-106, page 3, and recognize in limitation section in lines 336-342, pages 12-13.

Point 4: In the Data Analysis section, can you include a statement on how you managed missing data or incomplete questionnaires? If there were no missing data or incomplete questionnaire, then confirm this.

Response 4: Thank you for your comment. No incomplete data were recorded in our survey, as one of the exclusion criteria established that all responses must be mandatory. This measure was implemented to ensure the integrity of the collected data and the validity of the study results. We have included a comment in lines 104-106, page 3.

Point 5: In the Ethics section – I can see the ethics statement at the end of the work, but I was not clear regarding whether or not you formally gained ethical approval for the study, Could you please confirm within the text?

Response 5:  Thank you for your highlight. Following the submission of the manuscript and at the request of the Managing Editor, both the Institutional Review Board Statement and the Informed Consent Statement were rewritten in lines 374-398, in pages 13-14. 

Point 6: Results: The tables are well laid out and very clear. The discussion was relevant to the results and linked to the wider literature.

Response 6: Thank you for your comment

Point 7: Editing errors: Title – Sectional

Response 7: Thank you for your comment. Amended in Title.

Point 8: Line 276/277: check sentence structure. Perhaps include ‘be’ – to practice autonomously, be women-centered…

Response 8: Thank you for your comment. Amended in line 283 in page 11.